# A Novel Method for Estimating Pitch and Yaw of Rotating Projectiles Based on Dynamic Constraints

**DOI:** 10.3390/s19235096

**Published:** 2019-11-21

**Authors:** Liangliang An, Liangming Wang, Ning Liu, Jian Fu, Yang Zhong

**Affiliations:** 1School of Energy and Power Engineering, Nanjing University of Science & Technology, Nanjing 210094, China; anliangno1@126.com (L.A.); fujian@njust.edu.cn (J.F.); fishing0508@163.com (Y.Z.); 2Beijing Key Laboratory of High Dynamic Navigation Technology, Beijing Information Science & Technological University, Beijing 100101, China; ning.liu@bistu.edu.cn

**Keywords:** rotating projectile, dynamics constraint, zero-crossing method, attitude measurement, geomagnetic azimuth

## Abstract

This paper addresses the difficult problem of measuring the attitude of a high-spinning projectile and presents a novel method for estimating the pitch and yaw angles of the projectile in flight. The method is based on analysis of the external moment of the rotating projectile during flight and theoretical derivations obtained from the dynamics’ equations. First, the principle of zero-crossing method is introduced, which explains the process of geomagnetic azimuth and roll measurements by the non-orthogonal geomagnetic sensor combination. Then, the dynamics constraint equations between the Euler angles and flight-path angle, trajectory deflection angle of the projectile are derived using the dynamics equations of the projectile rotating around the centroid, and analysis of the flight characteristics of the projectile in stable flight. Next, the spatial orientation relationship between pitch, yaw angles and magnetic azimuth is established based on the physical principle of geomagnetic azimuth. Finally, the pitch and yaw angles are estimated using the unscented Kalman filter (UKF), with the dynamics constraint equations serving as the driving equations. In the UKF prediction stage, the Runge-Kutta method is used to discretize the state equation that improves the prediction accuracy. Simulation results show that the proposed method can be used to accurately calculate the pitch and yaw angles, and results of experimental data processing also verify the feasibility of the proposed method for real-world applications.

## 1. Introduction

Due to ever-increasing accuracy requirements for precision-guided weapons, acquisition of accurate flight attitude information of projectiles has become crucially important for analyzing their flight dynamics, as well as providing support for the navigation & guidance system. At present, the most commonly used attitude measurement methods rely on solar sensors [1,2], angular rate gyros [3,4,5], inertial measurement units (IMU) [6,7,8] and magnetometers [9,10,11]. Among these methods, the solar sensors work effectively only under good weather conditions, the angular velocity gyros have an upper limit on the rotational speed of the projectile, and the IMU suffer from error accumulation. Therefore, for measuring attitude of high-speed rotating projectiles, special working conditions, i.e., high temperature, high pressure, high overload, and high speed, as well as the requirements of low cost and small size preclude the use of many sensors. The magnetometer can be widely used in attitude estimation of rotating objects [12,13,14,15,16,17,18] after undergoing a calibration and compensation process [19,20,21], thanks to its features of reliable performance, low cost, and no error accumulation.

Many domestic and international scholars have carried out research on measuring the attitude of rotating projectiles using magnetometers. Wilson M. described several attitude measurement solutions based on combinations of multiple low-cost sensors [22] and conducted an in-depth research on the application of magnetometers in smart bomb [23]. Changey S. et al. conducted lab and flight experiments to verify the effectiveness of an algorithm that estimates roll of the projectile based on data acquired by two magnetometers [24,25]. Maley J. proposed a full attitude estimation method for spin-stabilized projectiles based on steady-state Kalman filtering [26]. Rogers J. et al. developed a low-cost orientation estimator for smart bombs equipped with magnetometers and thermopiles. This orientation estimator, with the advantage of not relying on GPS and other state feedbacks, estimates Euler angles and rotation rates using extended Kalman filtering (EKF) [27]. Most of these conventional methods calculate the Euler angles using the transfer matrix between the relevant coordinate systems [28,29]. Since the solution of the three Euler angles is non-independent, data fusion with devices such as built-in sensors and thermopiles is required in the calculation process [30,31,32]. However, the methods dependent on other built-in devices require sufficient internal space and specific working conditions. Apart from increasing production cost, adding extra electronic devices in a fixed space also increases the errors and noise of the output signal. Researchers in the United States first proposed a method for measuring the magnetic azimuth and rotational speed of a rotating projectile using two uniaxial magnetometers with a specific angle to the projectile axis [33,34]. This low cost and high precision method is called the ‘zero-crossing method’. Based on this method, experts at the Nanjing University of Science and Technology proposed an extreme value ratio method that combined non-orthogonal magnetic sensors, and conducted research in related areas [35,36]. Researchers at the Beijing Information Science and Technology University proposed a novel phase shift ratio method based on the extreme value ratio method [37]. In this study, a novel technique was developed based on the zero-crossing method. The moment applied to the rotating projectile flying in the air was analyzed and then the angular relationship contained in the external moment was extracted based on the ballistic characteristics of the projectile in stable flight. Subsequently, the constraint relationship between pitch, yaw angles and flight-path angle, trajectory deflection angle were deduced.

The EKF has been used in several studies for attitude estimation [38,39,40]. As it requires linearization of a nonlinear system before performing Kalman filtering, it is suitable for linear or weakly nonlinear systems. The EKF also has stringent requirements on the accuracy of the filter parameters, and involves calculation of the Jacobian matrix that is cumbersome. Compared with the EKF, the unscented Kalman filter (UKF) exhibits good robustness in the presence of nonlinearity and uncertainty [41], therefore, it is better at dealing with complex models with high nonlinearity [42,43,44,45,46] and has been used widely in recent years. It is necessary to discretize a continuous system when the filtering algorithm is applied in computers, and the discretization method and discrete step-size directly affect the filtering accuracy. When the step-size is large, the discrete models processed by the conventional methods such as the Euler method significantly differ from the continuous models. On the other hand, reducing the discrete step-size increases the computational complexity. When the fourth-order classical Runge-Kutta method [47,48,49] is used as the discretization method, the reliance on discrete step-size is reduced greatly. Consequently, the discrete models become closer to the theoretical continuous models, and the filtering accuracy is improved.

The method proposed in this paper works as follows: First, the dynamic constraint equations between pitch, yaw angles and flight-path angle, trajectory deflection angle are derived and used as the state model. Then, the geomagnetic vector and the projectile axis vector are simultaneously projected onto the reference coordinate system to obtain the spatial orientation relationship between the pitch, yaw angles and magnetic azimuth, and a measurement model based on geomagnetic azimuth is constructed. Finally, the pitch and yaw angles of the rotating projectile are estimated using the UKF algorithm, which utilizes the fourth-order classical Runge-Kutta method as the discretization method. The effectiveness of the proposed method is verified through simulations and processing of experimental data.

## 2. Definition of Coordinate System and Principle of Zero-Crossing Method

### 2.1. Coordinate Systems

To establish the differential equations of projectile dynamics, we use the approach described in [50] to introduce several basic coordinate systems: the reference coordinate system O−XYZ, the ballistic coordinate system $O-X_{2}Y_{2}Z_{2}$, the projectile axis coordinate system O−ξηζ and the second projectile axis coordinate system $O-\xi \eta_{2}\zeta_{2}$. Figure 1 shows the angular relationships between these coordinate systems. In the figure, the angles θ and ψ are the Euler angles of the pitch and yaw, respectively, the angle $\theta_{a}$ is the angle between the velocity vector and the horizontal plane, the angle $\psi_{2}$ is the angle between the velocity vector and the vertical plane, respectively, i.e., flight-path angle and trajectory deflection angle, and δ is the total attack angle of the projectile. Figure 2 further illustrates the pitch component $\delta_{1}$ and the yaw component $\delta_{2}$ of the total attack angle.

Both O−ξηζ and $O-\xi \eta_{2}\zeta_{2}$ are non-rolling coordinate systems that do not roll with the projectile. The axis Oξ of each coordinate system is the vertical axis of the projectile and the only difference between the coordinate planes Oηζ and $O\eta_{2}\zeta_{2}$ is a turning angle β [50].

### 2.2. Principle of Zero-Crossing Method

When a projectile and a uniaxial magnetometer with an angle of λ to the projectile axis rotate together in the earth’s magnetic field, the instantaneous field strength along the sensitive axis of the magnetometer is as follows [34]:(1)$$M_{S}=cos(\lambda )\left|\vec{M}\right|cos(\sigma_{M})+sin(\lambda )\left|\vec{M}\right|sin(\sigma_{M})sin(\phi )$$
where $\vec{M}$ is the geomagnetic field vector, $\sigma_{M}$ is the magnetic azimuth, i.e., the angle between the projectile axis and the direction of geomagnetic field, ϕ is the roll of the projectile and λ is the angle between the sensitive axis of the uniaxial magnetometer and the projectile axis. The rotating projectile is considered to be flying steadily in the air, and the output signal of the magnetometer changes periodically. When the sensitive axis is orthogonal to the direction of the earth’s magnetic field, the output signal of the magnetometer is zero, and the roll phase of the projectile represents the zero-crossing. There are two zero-crossings in a single cycle.

The zero-crossing method uses two uniaxial magnetometers ($S_{1}$ and $S_{2}$) with different mounting angles, as shown in Figure 3. With the mounting angles of 90° and 60°, the two magnetometers are in a coplanar relation with the projectile axis, i.e., they have equal initial roll phases. Four zero crossings can be extracted using the output signals of the two magnetometers, which results in two pairs of rolls given as ($\varphi_{S_{1A}}$,$\varphi_{S_{1B}}$) and ($\varphi_{S_{2A}}$,$\varphi_{S_{2B}}$). The ratio Φ can be calculated as
(2)$$\Phi =\left|\frac{\varphi_{S_{2B}}-\varphi_{S_{2A}}}{\varphi_{S_{1B}}-\varphi_{S_{1A}}}\right|$$

The magnetic azimuth of the projectile relative to the geomagnetic field during the flight can be determined based on the magnetic azimuth-ratio diagram plotted beforehand, and the roll angular rate and the roll phase angle of the projectile can be obtained using the recorded zero-crossing time.

The three-element attitude information, i.e., roll, pitch, and yaw angles, is converted into two-element attitude information, i.e., roll angle and geomagnetic azimuth. Subsequently, the roll angle information is separated to provide the possibility for secondary processing of the pitch and yaw angles, which is a strength of the zero-crossing method.

## 3. Method for Estimating Pitch and Yaw

The movement of the projectile in air consists of two parts: the centroid motion and the around- centroid motion. The former is mainly characterized by the position and velocity of the projectile, and is governed by the law of centroid movement. The latter is characterized by the attitude of the projectile, and is governed by the theorem of angular momentum [50].

### 3.1. Dynamics Constraint Equations

It is necessary to analyze the moment of external forces relative to the center of mass during the flight of the projectile. When there is no wind and the projectile shape does not cause any aerodynamic eccentricity, only the static and equatorial damping moments need to be considered. References [50,51] provide the dynamics equations of the projectile undergoing around-centroid motion. A new set of dynamic equations based on the specific problem are obtained as follows:(3)$$\{\begin{array}{l} \dot{\omega_{\eta }}=\frac{1}{A}M_{\eta }-\frac{C}{A}\omega_{\xi }\omega_{\zeta }\\ \dot{\omega_{\zeta }}=\frac{1}{A}M_{\zeta }+\frac{C}{A}\omega_{\xi }\omega_{\eta }\\ \dot{\theta }=\frac{\omega_{\zeta }}{cos(\psi )}\\ \dot{\psi }=-\omega_{\eta }\\ \dot{m_{z}^{\prime }}=0\\ \dot{m_{zz}^{\prime }}=0 \end{array}$$
where $M_{\eta }$ and $M_{\zeta }$ are components of the external moments in the projectile axis coordinate system O−ξηζ, A and C are coefficients of the moment of inertia, $\omega_{\xi }$, $\omega_{\eta }$ and $\omega_{\zeta }$ are projection components of the angular velocity on coordinate system O−ξηζ and $m_{z}^{\prime }$ and $m_{zz}^{\prime }$ are derivatives of the static and equatorial damping moment coefficients.

The external moments include static and equatorial damping moments, and their vector forms are
(4)$${\vec{M}}_{z}=\frac{\rho Sl}{2}v_{r}m_{z}\frac{1}{sin\delta_{r}}\left({\vec{v}}_{r}\times \vec{\xi }\right)$$
(5)$${\vec{M}}_{zz}=-\rho v_{r}Sldm_{zz}^{\prime }\vec{\omega }/2$$
where ${\vec{M}}_{z}$ and ${\vec{M}}_{\mathrm{zz}}$ are the static moment vector and equatorial damping moment vector, ρ is the air density, S is the cross-sectional area of the projectile, l is the projectile length, d is the projectile diameter, ${\vec{v}}_{r}$ is the velocity vector of the projectile relative to the wind, $m_{z}$ is the static moment coefficient, $\delta_{r}$ is the relative attack angle, $\vec{\xi }$ is the unit vector of the axis Oξ of the coordinate system O−ξηζ and $\vec{\omega }$ is the projectile oscillation angular velocity. When there is no wind, ${\vec{v}}_{r}$ is equal to $\vec{v}$, and $\delta_{r}$ is the attack angle δ.

For a small attack angle, $m_{z}=m_{z}^{\prime }\delta_{r}$, and the form of the static moment vector in Equation (4) can be rewritten as follows:(6)$${\vec{M}}_{z}=\frac{\rho Sl}{2}v_{r}m_{z}^{\prime }\left({\vec{v}}_{r}\times \vec{\xi }\right)$$

The component form of the static moment in the projectile axis coordinate system O−ξηζ is given as
(7)$$\begin{array}{l} M_{z\xi }=0\\ M_{z\eta }=\frac{\rho Sl}{2}v_{r}m_{z}^{\prime }v_{r\zeta }\\ M_{z\zeta }=-\frac{\rho Sl}{2}v_{r}m_{z}^{\prime }v_{r\eta } \end{array}$$
where $v_{r\eta }$ and $v_{r\zeta }$ are components of the relative velocity ${\vec{v}}_{r}$ in the coordinate system O−ξηζ. Let the components of the relative velocity ${\vec{v}}_{r}$ in the coordinate system $O-\xi \eta_{2}\zeta_{2}$ be denoted as $v_{r\eta_{2}}$ and $v_{r\zeta_{2}}$, the relationship between two components is as follows:(8)$$\begin{array}{l} v_{r\eta }=v_{r\eta_{2}}cos\beta +v_{r\zeta_{2}}sin\beta \\ v_{r\zeta }=-v_{r\eta_{2}}sin\beta +v_{r\zeta_{2}}cos\beta \end{array}$$

For a normally flying projectile, as the attack angle and the ballistic deflection are small, $\delta_{1}$, $\delta_{2}$, ψ, $\psi_{2}$ and $\theta -\theta_{a}$ have small values. Thus, the following relationship holds [50]:(9)$$\beta \approx 0;\;\delta_{1}\approx \theta -\theta_{a};\;\delta_{2}\approx \psi -\psi_{2}$$

As shown in Figure 2, the rotation relationship between the ballistic coordinate system $O-X_{2}Y_{2}Z_{2}$ and the second projectile axis coordinate system $O-\xi \eta_{2}\zeta_{2}$ leads to $v_{r\eta_{2}}=-v\delta_{1}$ and $v_{r\zeta_{2}}=-v\delta_{2}$. Consequently, Equation (8) can be further written as
(10)$$\begin{array}{l} v_{r\eta }=v_{r\eta_{2}}=-v\delta_{1}\\ v_{r\zeta }=v_{r\zeta_{2}}=-v\delta_{2} \end{array}$$

If the influence of wind is ignored, the static moment components of the Oη and Oζ axes in Equation (7) can be written as
(11)$$\begin{array}{l} M_{z\eta }=\frac{\rho Sl}{2}vm_{z}^{\prime }\left(-v\delta_{2}\right)\\ M_{z\zeta }=-\frac{\rho Sl}{2}vm_{z}^{\prime }\left(-v\delta_{1}\right) \end{array}$$

Substituting Equation (9) into the above equation, we obtain
(12)$$\begin{array}{l} M_{z\eta }=-\frac{\rho Sl}{2}v^{2}m_{z}^{\prime }\left(\psi -\psi_{2}\right)\\ M_{z\zeta }=\frac{\rho Sl}{2}v^{2}m_{z}^{\prime }\left(\theta -\theta_{a}\right) \end{array}$$

Defining $A_{m}=\frac{\rho Sl}{2A}m_{z}^{\prime }$, the static moment components $M_{z\eta }$ and $M_{z\zeta }$ can be rewritten as
(13)$$\begin{array}{l} M_{z\eta }=-AA_{m}v^{2}\left(\psi -\psi_{2}\right)\\ M_{z\zeta }=AA_{m}v^{2}\left(\theta -\theta_{a}\right) \end{array}$$

In the same way, the equatorial damping moment in Equation (5) can be written in component form in the coordinate system O−ξηζ as follows:(14)$$\begin{array}{l} M_{zz\xi }=-\frac{\rho v_{r}}{2}Sldm_{zz}^{\prime }\omega_{\xi }\approx 0\\ M_{zz\eta }=-\frac{\rho v_{r}}{2}Sldm_{zz}^{\prime }\omega_{\eta }\\ M_{zz\zeta }=-\frac{\rho v_{r}}{2}Sldm_{zz}^{\prime }\omega_{\zeta } \end{array}$$

Similarly, under the condition of no wind, defining $C_{m}=-\frac{\rho Sld}{2A}m_{zz}^{\prime }$, the components $M_{zz\eta }$ and $M_{zz\zeta }$ of the equator damping moment can be rewritten as
(15)$$\begin{array}{l} M_{zz\eta }=AC_{m}v\omega_{\eta }\\ M_{zz\zeta }=AC_{m}v\omega_{\zeta } \end{array}$$

Considering both Equations (13) and (15), the components $M_{\eta }$ and $M_{\zeta }$ of the total external moment can be rewritten as
(16)$$\begin{array}{l} M_{\eta }=M_{z\eta }+M_{zz\eta }=-AA_{m}\upsilon^{2}\left(\psi -\psi_{2}\right)+AC_{m}\upsilon \omega_{\eta }\\ M_{\zeta }=M_{z\zeta }+M_{zz\zeta }=-AA_{m}\upsilon^{2}\left(\vartheta -\theta_{a}\right)+AC_{m}\upsilon \omega_{\zeta } \end{array}$$

Substituting Equation (16) into (3), the dynamics constraint equations including flight-path angle, trajectory deflection angle and two Euler angles are obtained as follows:(17)$$\{\begin{array}{l} \dot{\omega_{\eta }}=-A_{m}v^{2}\left(\psi -\psi_{2}\right)+C_{m}v\omega_{\eta }-\frac{C}{A}\omega_{\xi }\omega_{\zeta }\\ \dot{\omega_{\zeta }}=A_{m}v^{2}\left(\vartheta -\theta_{a}\right)+C_{m}v\omega_{\zeta }+\frac{C}{A}\omega_{\xi }\omega_{\eta }\\ \dot{\vartheta }=\frac{\omega_{\zeta }}{\cos(\psi )}\\ \dot{\psi }=-\omega_{\eta }\\ \dot{A_{m}}=0\\ \dot{C_{m}}=0 \end{array}$$
where the projectile’s flight speed v, flight-path angle $\theta_{a}$ and trajectory deflection angle $\psi_{2}$ can be calculated using the ballistic radar data. The axial angular velocity of the projectile is denoted by $\omega_{\xi }$.

### 3.2. Relationship between Pitch, Yaw Angles and Magnetic Azimuth

When the projectile is flying in the air, its instantaneous attitude relative to the earth’s magnetic field can be represented by the pitch, yaw, magnetic dip and magnetic declination. Since the projectile’s position information can be detected by the radar, the geomagnetic field information, including magnetic dip and magnetic declination of the projectile’s position can be calculated based on the geomagnetic field model. Therefore, the magnetic azimuth $\sigma_{M}$ only contains two pieces of unknown information, i.e., the pitch and yaw.

The shooting direction is denoted as $\alpha_{N}$, and the influence of the meridional convergence angle is ignored. The unit vectors on the three axes of the reference coordinate system O−XYZ are denoted as $\vec{i}$, $\vec{j}$ and $\vec{k}$. The geomagnetic field vector is projected onto the reference coordinate system. 

As shown in Figure 4, the geomagnetic vector is described by the north-east-down (NED) coordinate system. Taking the northern hemisphere as an example, the geomagnetic unit vector and its horizontal projection are $\vec{M}$ and ${\vec{M}}_{N}$, respectively, the magnetic declination is D, north to the east is positive, the magnetic dip is I and the downward direction is positive.

Thus, the geomagnetic unit vector can be expressed as
(18)$$\vec{M}=cos(I)cos(a_{N}-D)\vec{i}-sin(I)\vec{j}-cos(I)sin(a_{N}-D)\vec{k}$$
where both the magnetic declination D and the magnetic dip I can be calculated based on the spherical harmonics model of the geomagnetic field, and the shooting direction $\alpha_{N}$ is known before the experiment.

The projectile axis unit vector $\vec{\xi }$ can be obtained by projecting the projectile axis vector onto the reference coordinate system O−XYZ as shown in Figure 1.
(19)$$\vec{\xi }=cos(\theta )cos(\psi )\vec{i}+sin(\theta )\vec{j}+cos(\theta )sin(\psi )\vec{k}$$

The magnetic azimuth $\sigma_{M}$ is the angle between the geomagnetic unit vector and the first projectile axis unit vector. It can be calculated as follows using the vector included angle cosine formula:(20)$$cos(\sigma_{M})=\frac{\vec{M}\cdot \vec{\xi }}{\left|\vec{M}\right|\left|\vec{\xi }\right|}=cos(I)cos(\alpha_{N}-D)cos(\theta )cos(\psi )\;-sin(I)sin(\theta )-cos(I)sin(\alpha_{N}-D)cos(\theta )sin(\psi )$$

### 3.3. Estimation of Dip and Yaw

When the derived dynamics constraint equations are used as the driving equations, the pitch and yaw angles of the rotating projectile can be estimated based on the spatial relationship between the magnetic azimuth and the Euler angles. A block diagram of the attitude estimation method is shown in Figure 5. The built-in magnetometer provides accurate geomagnetic signals. The magnetic azimuth and rotational speed serve as the inputs to the estimation algorithm and can be calculated using the zero-crossing method. The rotational speed can be used to obtain the roll. The radar collects the velocity and position information, and calculates the geomagnetic field information of the entire trajectory based on the geomagnetic field model to provide support to filtering. The initial firing elements are used to simulate the magnetic azimuth of the initial section of the trajectory and perform initial filtering calibration using the calculated magnetic azimuth as a reference. Finally, the pitch and yaw angles are estimated using the improved UKF algorithm.

## 4. Design of UKF

The UKF mainly consists of two phases: the prediction phase and the correction phase. In the prediction phase, a set of state prediction points based on the sigma points should be generated. Since the state equation is a continuous model, discretization needs to be carried out that can directly affect the accuracy of the filtering results. The Runge-Kutta method is often used in ballistic calculations as it outperforms other methods in terms of discretization accuracy under the same step-size. In this paper, the fourth-order classical Runge-Kutta method is used for state estimation in the prediction stage. Therefore, the filtering algorithm used in this paper is called the RK4-UKF algorithm.

Assume that the state equation of a continuous nonlinear system is
(21)$${\dot{X}}_{k}=f\left[X_{k-1},k-1\right]+W_{k-1}$$

The measurement equation is
(22)$$Y_{k}=h\left[X_{k},k\right]+V_{k}$$

The workflow of the RK4-UKF algorithm is as follows:
Calculation of the sigma point set
(23)$$X_{k-1}^{0}={\hat{x}}_{k-1};\;X_{k-1}^{i}=\{\begin{array}{ll} {\hat{x}}_{k-1}+\sqrt{\left(n+\lambda \right)P_{x}} & i=1,2,\ldots ,n\\ {\hat{x}}_{k-1}-\sqrt{\left(n+\lambda \right)P_{x}} & i=n+1,\ldots ,2n \end{array}$$Prediction phase
(24)$$\begin{array}{l} k_{1}=f\left(X_{k-1}^{i}\right);\;k_{2}=f\left(X_{k-1}^{i}+\frac{h}{2}k_{1}\right);\;k_{3}=f\left(X_{k-1}^{i}+\frac{h}{2}k_{2}\right);\;k_{4}=f\left(X_{k-1}^{i}+hk_{3}\right)\\ X_{k/k-1}^{i}=X_{k-1}^{i}+\frac{h}{6}\left(k_{1}+2k_{2}+2k_{3}+k_{4}\right)\\ {\hat{x}}_{k/k-1}=\overset{2n}{\underset{i=0}{\sum }}W_{i}^{m}X_{k/k-1}^{i};\;P_{k/k-1}=\overset{2n}{\underset{i=0}{\sum }}W_{i}^{c}\left[X_{k/k-1}^{i}-{\hat{x}}_{k/k-1}\right]{\left[X_{k/k-1}^{i}-{\hat{x}}_{k/k-1}\right]}^{T}+Q_{k} \end{array}$$Correction phase
(25)$$Y{\;}_{k/k-1}^{i}=h\left(X_{k/k-1}^{i}\right);\;{\hat{y}}_{k/k-1}=\overset{2n}{\underset{i=0}{\sum }}W_{i}^{m}Y{\;}_{k/k-1}^{i}$$
(26)$$P_{\left(YY\right)k/k-1}=\overset{2n}{\underset{i=0}{\sum }}W_{i}^{c}\left[Y{\;}_{k/k-1}^{i}-{\hat{y}}_{k/k-1}\right]{\left[Y{\;}_{k/k-1}^{i}-{\hat{y}}_{k/k-1}\right]}^{T}+R_{k}$$
(27)$$P_{\left(XY\right)k/k-1}=\overset{2n}{\underset{i=0}{\sum }}W_{i}^{c}\left[X_{k/k-1}^{i}-{\hat{x}}_{k/k-1}\right]{\left[Y_{k/k-1}^{i}-{\hat{y}}_{k/k-1}\right]}^{T}$$
(28)$$K_{k}=P_{\left(XY\right)k/k-1}P_{\left(YY\right)k/k-1}^{-1}$$
(29)$${\hat{X}}_{k}={\hat{X}}_{k/k-1}+K_{k}\left(Y_{k}-{\hat{y}}_{k/k-1}\right)$$
(30)$$P_{k}=P_{k/k-1}-K_{k}P_{\left(YY\right)k/k-1}K_{k}^{T}$$

### 4.1. State Equation

Given the continuous nonlinear state equations in Equation (17), the state variables are written as
(31)$$x=\left[x_{1}\;x_{2}\;x_{3}\;x_{4}\;x_{5}\;x_{6}]=[\omega_{\eta }\;\omega_{\zeta }\;\theta \;\psi \;A_{m}\;C_{m}\right]$$

Then, Equation (17) can be written as
$$\dot{X}=f\left(x\right)=\left(\begin{array}{c} -x_{5}v^{2}\left(x_{4}-\psi_{2}\right)+x_{1}x_{6}-\frac{C}{A}\omega_{\xi }x_{2}\\ x_{5}v^{2}\left(x_{3}-\theta_{a}\right)+x_{2}x_{6}+\frac{C}{A}\omega_{\xi }x_{1}\\ \frac{x_{2}}{cosx_{4}}\\ -x_{1}\\ 0\\ 0 \end{array}\right)+W$$

As the nonlinear equations given in Equation (31) only approximately describe the around-centroid motion of the projectile, there will be certain errors. Therefore, Gaussian white noise W∼N(0,Q) is introduced to model these errors.

### 4.2. Measurement Equation

The magnetic azimuth is represented by a measured variable $y=\left(\sigma_{M}\right)$. The measurement equation can be constructed as follows, based on Equation (20):
(32)$$\begin{array}{ll} y & =h\left(x\right)+V\\ & =\arccos(cos(I)cos(\alpha_{N}-D)cos(x_{3})cos(x_{4})-sin(I)sin(x_{3})-cos(I)sin(\alpha_{N}\;-D)cos(x_{3})sin(x_{4}))+V \end{array}$$
where measurement noise V is the Gaussian white noise, given as V∼N(0,R), and $R=\left(\sigma_{\sigma_{M}}^{2}\right)$.

## 5. Simulation and Experimental Results

### 5.1. Simulation and Analysis

#### 5.1.1. Simulation

The calculation steps for the magnetic azimuth and roll are described in detail in [35]. Therefore, these steps will not be repeated here and instead, only the simulation results will be given. The focus of simulations in this study is the estimation of pitch and yaw angles. Assume that the projectile is launched from a location of (E100°, N39°) with an initial velocity of 800 m/s, a shooting angle of 60° and a shooting direction of 100°. The pitch and yaw components of the angular velocity equal to 2 rad/s are added to simulate the initial disturbance at the time of launch, and the ballistic data are simulated using the 6D ballistic equations. Then, the geomagnetic signal output information of the trajectory is simulated through conversion between the relevant coordinate systems. Finally, the magnetic azimuth is calculated using the zero-crossing method and serves as the true value. 

A Gaussian white noise *d* ~ *N*(0,0.5°) is added to the true value of the magnetic azimuth to serve as measurement value. Figure 6a shows the simulated and true values of the initial 1 s of the ballistics, and Figure 6b shows the discrepancy between the true and simulated measured values. It can be observed that the maximum error of the simulated measured value is about ±1.6°, which is much larger than the measurement error described in [34].

The pitch angle and yaw angle are estimated using the RK4-UKF algorithm and compared with the corresponding true values, as shown in Figure 7. Figure 7a shows the estimation of the pitch and yaw angles of the entire ballistic. It can be seen from the figure that the projectile flied for more than 100 s. The estimated value is close to the true value in the entire ballistic, and the estimated effect is satisfactory. From the law of yaw angle movement, it can be seen that the existence of dynamic equilibrium angle causes the yaw angle to deviate to the right from the trajectory deflection angle (the yaw angle was defined as positive to the right, so the yaw angle is positive) in the midcourse because of the right-hand twist of projectile. Figure 7b shows the estimation error of the pitch and yaw angles, both of which are mostly in the range of (−0.2° ~ 0.2°). The errors are slightly larger in the beginning of the trajectory phase, and fluctuate in the midcourse. Figure 7c,d show the estimated and true values of the pitch and yaw angles within 1 s of the initial phase of ballistic. The dual-circular motion law of the projectile can be seen clearly from the figures. The pitch and yaw angles oscillate periodically around flight-path angle and trajectory deflection angle, respectively. This oscillation is a slow-circular motion, with a low frequency and continuously diminishing amplitude. At the same time, the projectile axis oscillates periodically around the dynamic balance axis in a fast-circular fashion, with a continuously diminishing nutation amplitude. 

To show movement of the body axis, the oscillation trajectory of the projectile is constructed based on the pitch and yaw angles, as shown in Figure 8. The projectile starts from a position of (60°,0°) and undergoes a counterclockwise dual-circular motion. The estimated oscillation trajectory of the projectile obtained using the RK4-UKF algorithm is consistent with the true trajectory. For the sake of clarity, only five cycles of the fast-circular motion in the initial phase and a slow circular motion cycle consisting of their centers are shown in the figure. Each red dot in the figure shows the approximate center of the fast-circular motion and the dotted line passing through the center represents the slow-circular motion, i.e., the motion trajectory of the dynamic balance axis of the projectile. The slow-circular motion is also in the counterclockwise direction. The dual-circular motion of the projectile is prominent.

#### 5.1.2. Monte Carlo Simulation

In order to avoid the contingency of the RK4-UKF simulation results, a Monte Carlo simulation was performed. The RK4-UKF was run 1000 times, and the initial value of the state variable and the noise of the measurement value were randomly changed within a reasonable range before each run. After 1000 runs, the error of pitch and yaw angles estimated by RK4-UKF are tested in terms of the mean, mean square error and maximum of the absolute value, respectively. At the same time, the normal curve fitting was performed on the Monte Carlo test results. The test results are shown in Figure 9.

For the estimation error of the pitch angle, the expectation of mean is about 9.587 × 10^−5^ degree, the expectation of mean square error is about 0.065°, and the expectation of maximum of the absolute value is about 0.2644°. For the estimation error of the yaw angle, the expectation of mean is about −1.675 × 10^−4^ degree, the expectation of mean square error is about 0.05984°, and the expectation of maximum of the absolute value is about 0.1985°. Figure 9c is the maximum of the absolute value of the estimation error, i.e., the maximum estimation error. According to the principle of normal distribution, the maximum estimation error of the pitch angle does not exceed 0.457°, and the maximum estimation error of the yaw angle does not exceed 0.297°.

#### 5.1.3. Analysis of simulation results

The simulation results show that the filtering result of the RK4-UKF algorithm is consistent with the true value, with a small error on the order of 10^−1^ degrees. The Monte Carlo simulation also shows that the pitch angle error estimated by this method does not exceed 0.457°, and the yaw angle error does not exceed 0.297°. The following conclusions were obtained based on the simulation results:
When analyzing the moment applied to the projectile during flight, it is assumed that the projectile shape has no eccentricity and there is no wind. Among external moments, only the static and equatorial damping moments are considered, while the smaller Magnus moment is ignored. Moreover, since the attack angle is small, the resulting small ballistic deviation from the firing surface allows approximations of $\delta_{1}\approx \theta -\theta_{a};\;\delta_{2}\approx \psi -\psi_{2}$ during the state equation derivation process. Thus, there is uncertainty in the adjustment of the state noise parameters.Under the same sampling step-size, the fourth-order classical Runge-Kutta discretization method results in smaller discretization errors compared to other method such as the Euler method, and its discrete equations are close to the continuous model.

### 5.2. Experiment and Analysis

#### 5.2.1. Experiment

It is impossible to observe directly and record the attitude of a flying projectile using current technology when the range reaches tens of kilometers or hundreds of kilometers. However, based on the pattern of projectile motion and the Lyapunov stability principle [50,51], the stability of the flying projectile can be maintained only when the following two conditions are met: a) The directions of the slow circular motions of the lateral attitude parameters including pitch and yaw are consistent with the velocity direction during the flight of projectile; b) The projectile axis undergoes periodic nutation around the velocity direction and the amplitude diminishes continuously. Therefore, it is feasible to verify the effectiveness of the attitude estimation method using the flight-path angle and trajectory deflection angle measured by a radar or a GPS device.

The experimental verification was conducted at a shooting range. During the experiment, the weather was good and windless, with the presence of a few clouds. The field layout of the verification experiment is shown in Figure 10. The reference coordinate system is the North-Up-East coordinate. The elevation angle $\theta_{0}$ was 15.3° and the direction of fire $\alpha_{N}$ was 103.3555°. A velocity radar was set up near the artillery location to measure the projectile velocity and an air balloon was launched to collect the meteorological data.

The measurement system used in the experiment consisted of a CPU, a magnetometer unit consisting of two single-axis magnetometers, a data acquisition unit, a signal processing unit, a communication unit, a power supply and other auxiliary unit. Figure 11 shows the block diagram of the measurement system. The geomagnetic unit collected the original voltage signal, the signal processing unit carried out signal conversion and processing, the CPU was responsible for signal processing, and the communication unit was used for transmitting and receiving instructions. Figure 12 shows the photos of the measurement system. The system was mounted inside the standard projectile to form the assembly, as shown in Figure 13a.

The assembly was mounted at the front section of a standard warhead. Its interior was fixed using solid glue and protected with a non-magnetic cover. This arrangement enabled it to withstand the strong impact and large overload during the launch stage, ensuring normal operation of the measurement components. The assembly was recycled after launching, as shown in Figure 13b.

The initial velocity of the projectile measured by the radar was 744.4 m/s. During the flight, the magnetometer recorded the variation of geomagnetic intensity in each axial direction, and the magnetic azimuth and rotational speed of the projectile were calculated using the zero-crossing method. The calculation results are shown in Figure 14. 

Figure 14a shows the variation of magnetic azimuth along the entire trajectory. In the initial section of the trajectory, the projectile’s oscillation amplitude is relatively large due to the influence of initial disturbances. Then, the oscillation amplitude decreases gradually. Towards the end section of the trajectory, the projectile begins to oscillate again, and the oscillation amplitude increases continuously. There are two reasons behind this phenomenon: First, as the rotational speed of the projectile decreases, the gyroscopic effect of the projectile’s rotation diminishes gradually. Consequently, the dynamic stability of the projectile reduces gradually, eventually causing oscillations. Second, the change in the projectile’s velocity from supersonic to subsonic also causes oscillations. Radar data show that the projectile’s velocity was equal to 338.69 m/s (about Mach 1) around 24 s, which is in the transonic region.

Figure 14b shows the variation of magnetic azimuth during the initial 2 s section of the trajectory. The initial value of the magnetic azimuth is 117.6°, the minimum and maximum values are 112.4° and 127° in the first cycle of the slow circular motion, respectively, and the oscillation amplitude is about 7.3°. The pattern of dual-circular motion of the projectile can also be clearly seen from the figure. The oscillation amplitude diminishes continuously irrespective of fast or slow circular motion.

Figure 15 shows the variation of rotational speed of the projectile along the entire trajectory. The initial rotational speed of the projectile is 1486 rad/s, which finally drops to about 900 rad/s. This speed drop is fast at first and then gradually slows down.

#### 5.2.2. Initial Alignment

Initial alignment needs to be performed at first to determine the initial value of the filter. To carry out this alignment, the theoretical trajectory is simulated using the 6D rigid body ballistic equations based on the initial firing elements. Then, the geomagnetic information of the entire trajectory is obtained through conversion between the relevant coordinate systems. With this information, the theoretical magnetic azimuth angle is calculated using the zero-crossing method and compared with the measured value. The initial firing conditions and moment coefficients should be adjusted continuously until the theoretical magnetic azimuth becomes roughly consistent with the measured value.

Among the initial launch conditions, the position and velocity of the projectile are provided by the radar, and the elevation angle and direction of fire are known in advance. The initial rotational speed $w_{\xi 0}$ can be calculated either using the zero-crossing method, or based on the initial velocity as follows [50,51]:(33)$$\omega_{\xi 0}=\frac{2\pi v_{0}}{\eta d}$$
where η the twist pitch of is rifling, d is the diameter of the projectile and $v_{0}$ is the initial velocity of the projectile after leaving the gun muzzle. Therefore, it is necessary to only adjust the initial angular velocity and moment coefficient of the pitch and yaw directions to obtain simulated magnetic azimuth curve that matches the measured magnetic azimuth curve, as shown in Figure 16.

#### 5.2.3. Estimation of Pitch and Yaw Using RK4-UKF

The pitch and yaw angles of the projectile were estimated using the designed RK4-UKF algorithm, and the estimated values were compared with the flight-path angle and trajectory deflection angle obtained from the radar data, as shown in Figure 17a. The flight time of the projectile is about 32.4 s. The estimated value of the pitch angle is consistent with the flight-path angle and decreases with the decrease of the ballistic trajectory. The estimated value of the yaw angle is consistent with the trajectory deflection angle. Under the external action, the deflection direction is right. The amplitude of the projectile axis is large at the initial phase of ballistic because of initial disturbance, then decreases continuously, and increases again and at the terminal phase. Figure 17b,c show the motion of the projectile attitude clearly. The slow circular motion components of the projectile’s lateral attitude parameters, i.e., the pitch and yaw angles, oscillate around flight-path angle and trajectory deflection angle, respectively with continuously diminishing amplitudes, which fits with the pattern of flight stability of the projectile.

For a rotating projectile flying steadily in the air, the following approximations can be made: The discrepancy between the calculated pitch θ and flight-path angle $\theta_{a}$ is taken as the pitch component $\delta_{1}$ of the attack angle, and the discrepancy between the yaw ψ and trajectory deflection angle $\psi_{2}$ is taken as the yaw component $\delta_{2}$ of the attack angle, as shown in Figure 18. These approximations are also helpful for measuring the attack angle of the projectile.

#### 5.2.4. Discussion on Experimental Results

Based on the analysis of experimental data and filtering results, the following issues and relevant conclusions were obtained:In a real-world scenario, the actual static and equatorial damping moment coefficients of the flying projectile often deviated from the theoretical values that were determined based on the projectile design. Therefore, it is necessary to adjust the theoretical moment coefficient when performing the initial alignment based on trajectory simulation.Oscillation is bound to happen during the descending section of the actual trajectory. The larger the shooting angle, the larger the oscillation amplitude, which is the well-known Mayevsky problem. By contrast, the end section of the theoretical trajectory is free of oscillation. This is because the motion of projectile axis is obtained based on the pure kinematics theory, which assumes that the moment of momentum vector coincides with the projectile axis. Therefore, there is no projectile axis swing problem during the end section of the theoretical trajectory. The oscillation phenomenon that occurs during the descending section of the actual trajectory can be attributed to two factors: (1) A reduction of the gyro stability factor due to decreased rotational speed; (2) The dramatic change of the aerodynamic load that causes the projectile to oscillate when the projectile flight speed is in the transonic region.The method for estimating pitch and yaw angels proposed in this paper is based on the constraints of dynamics equations of projectile. Through proper approximations, the relationship between the attitude and velocity angles can be determined, i.e., the slow-motion terms of the lateral attitude of the projectile are consistent with the velocity direction. This is also the basis for determining the rationality of the filtering results.Limited by the current attitude measurement technology and experimental conditions, the true value of the projectile attitude cannot be obtained in the field experiment, and the accuracy of the estimation cannot be quantified. The experiment is mainly to verify the feasibility of the method in practical engineering applications. The method is proven to be feasible and effective through the analysis of the flight stability of the projectile. The quantification of estimation error by designing the verification experiment is the focus of the next step in the future.

## 6. Conclusions

In this paper, a novel method for estimating the pitch and yaw angles of a flying projectile was developed. Based on the analysis of the flight characteristics and external moment of the rotating projectile in steady flight, the dynamics constraint equations of the lateral attitude Euler angles and the velocity angles were derived using the projectile dynamics equation without relying on conversions between the relevant coordinate systems. The relationships between the pitch, yaw, and magnetic azimuth were established based on the spatial vector relationship. Finally, the pitch and yaw angles were estimated using the RK4-UKF algorithm. The feasibility and effectiveness of the proposed method were verified using simulation and experimental results, and different issues arising in the simulations and experiments were analyzed and discussed. The following points are worth noting:Although the geomagnetic azimuth used in the proposed method was calculated using the zero-crossing method, any other method can also be used.As the proposed method deals with high-spinning projectiles in steady flight, the magnetic declination and dip during the projectile flight can be obtained in two ways: (1) Calculation based on the geomagnetic model; (2) Calculation using the measured launch location based on the assumption that the magnetic declination and dip are constant at each location.The object studied in this paper is the idealized projectile, which only considers the static moment and the equatorial damping moment, and assumes that there is no wind. The influence of the wind field model, Magnus moment and the moment caused by the shape asymmetry on the attitude of projectile will be considered in the future works, which makes the simulation model more accurate and improves the accuracy of the estimation.

The proposed method seeks to break through the dynamic characteristics of projectile and opens up new directions for developing attitude estimation methods. Catering to the need of developing intelligent bombs, it is expected to play an important role in the navigation and guidance of artillery shells and high-spinning rockets, and precision control of flying objects.

## Figures and Tables

**Figure 1 sensors-19-05096-f001:**
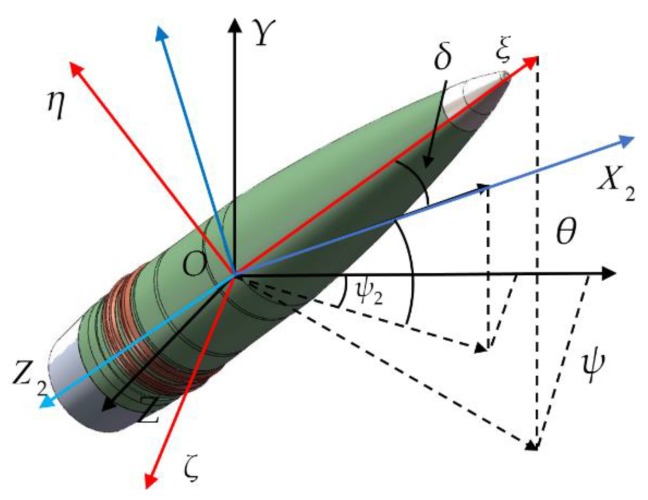
The angular relationships between these coordinate systems.

**Figure 2 sensors-19-05096-f002:**
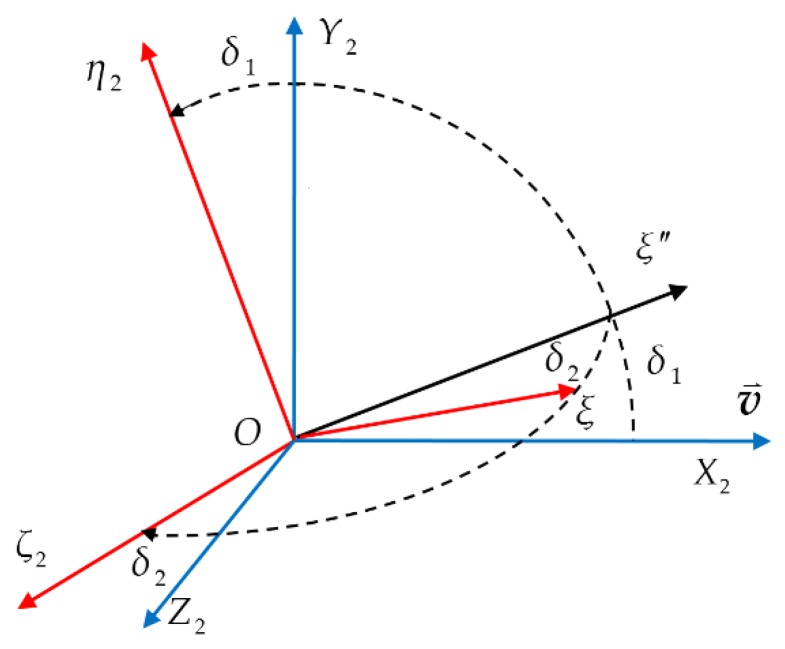
Coordinate system $O-X_{2}Y_{2}Z_{2}$ turns to Coordinate system $O-\xi \eta_{2}\zeta_{2}$.

**Figure 3 sensors-19-05096-f003:**
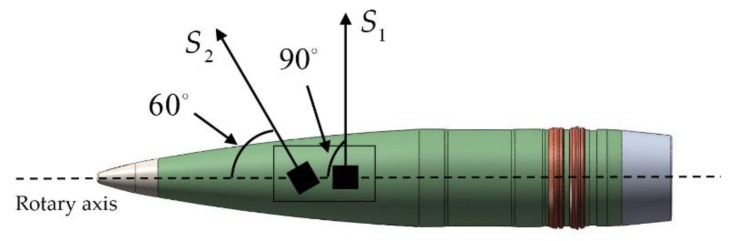
Installation diagram of two uniaxial magnetometers.

**Figure 4 sensors-19-05096-f004:**
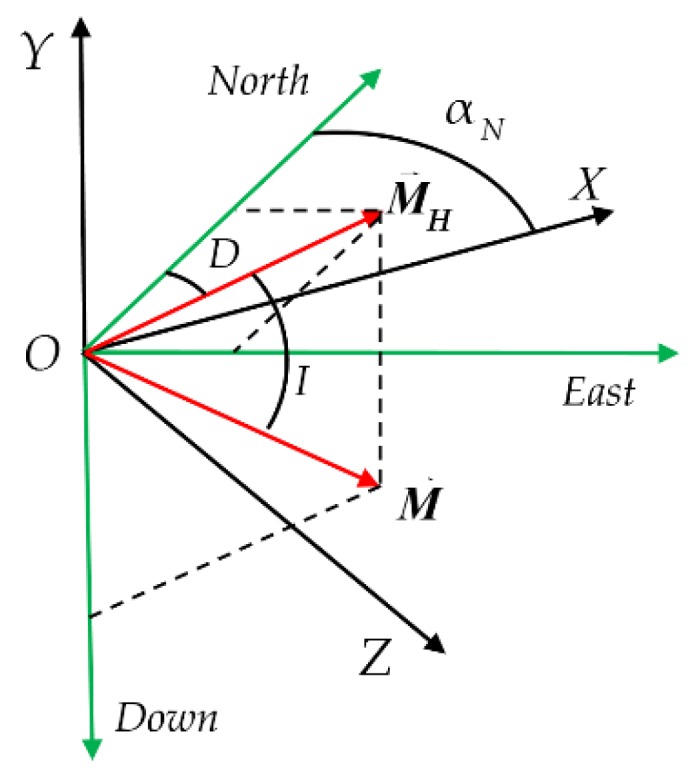
Orientation map of the reference coordinate system and geomagnetic vector.

**Figure 5 sensors-19-05096-f005:**
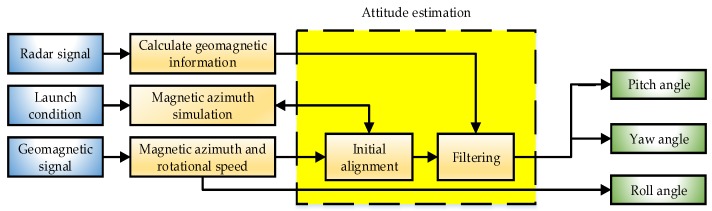
diagram of attitude estimation.

**Figure 6 sensors-19-05096-f006:**
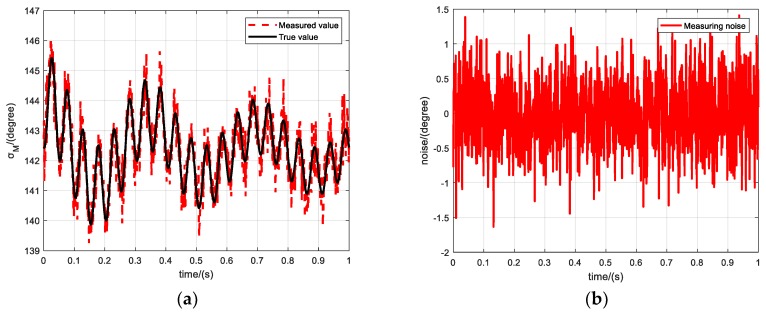
Simulation of magnetic azimuth measurement. (**a**) Magnetic azimuth measurement, (**b**) Measuring noise.

**Figure 7 sensors-19-05096-f007:**
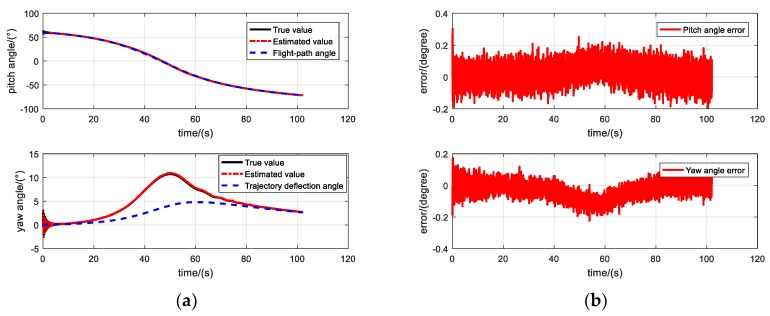
Estimated and true values of pitch and yaws. (**a**) Estimation of the pitch and yaw angles in the entire ballistic, (**b**) Estimation error of the pitch and yaw angles, (**c**) Estimation of the pitch angle within 1 s of the initial phase of ballistic, (**d**) Estimation of the yaw angle within 1 s of the initial phase of ballistic.

**Figure 8 sensors-19-05096-f008:**
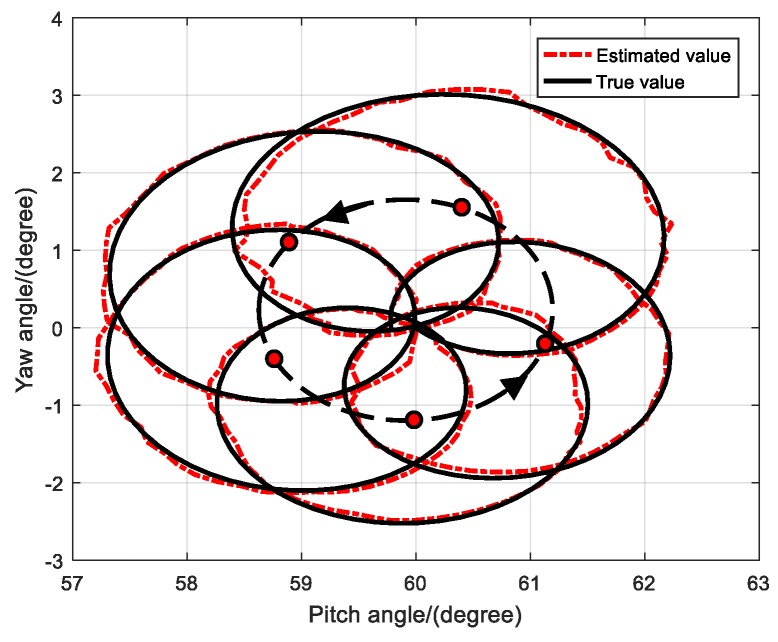
Oscillation trajectory of the projectile.

**Figure 9 sensors-19-05096-f009:**
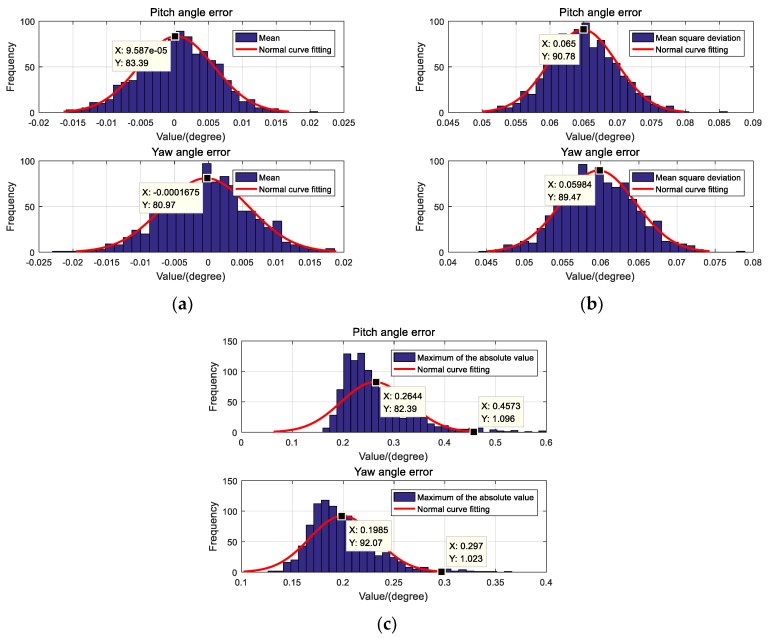
Results of Monte Carlo simulation. (**a**) Mean of estimation errors,(**b**) Mean square error of estimation errors,(**c**) Maximum of the absolute value of estimation errors.

**Figure 10 sensors-19-05096-f010:**
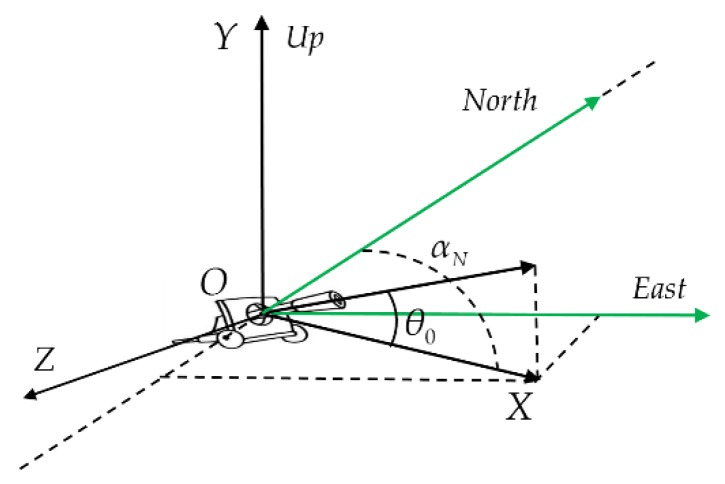
The launch angle and direction of the experiment.

**Figure 11 sensors-19-05096-f011:**
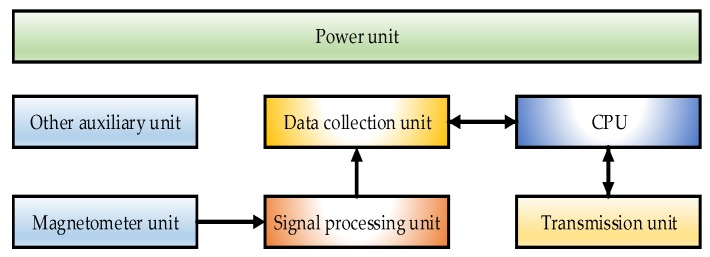
The block diagram of the measurement system.

**Figure 12 sensors-19-05096-f012:**
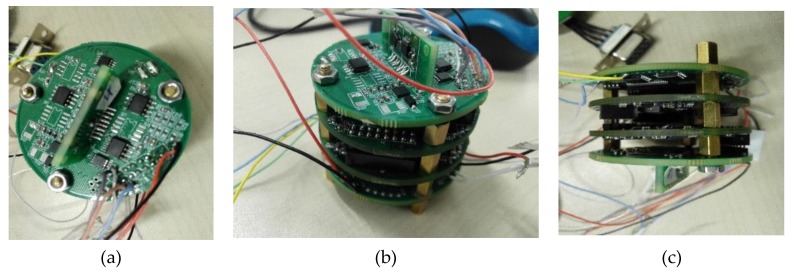
The photos of the measurement system. (**a**) Top view, (**b**) 45° view, (**c**) Side view

**Figure 13 sensors-19-05096-f013:**
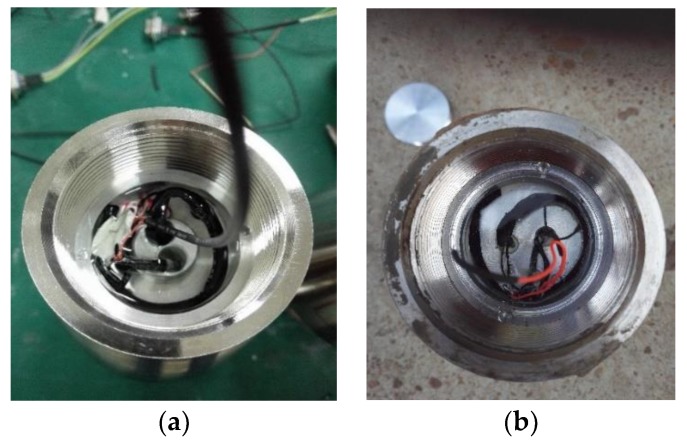
The photos of assembly. (**a)** Assembly before experiment, (**b**) Recycled assembly.

**Figure 14 sensors-19-05096-f014:**
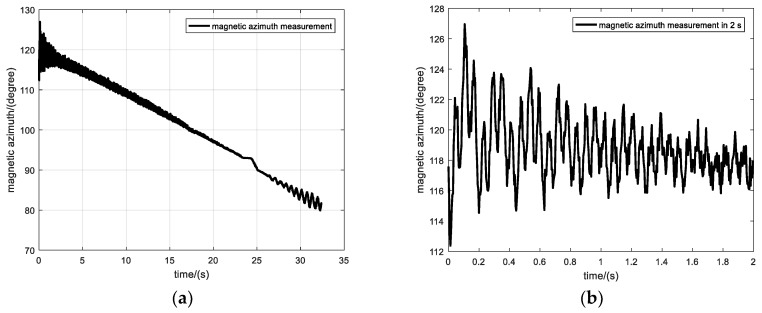
The magnetic azimuth calculated using the zero-crossing method. (**a**) Magnetic azimuth, (**b**) Magnetic azimuth in 1s.

**Figure 15 sensors-19-05096-f015:**
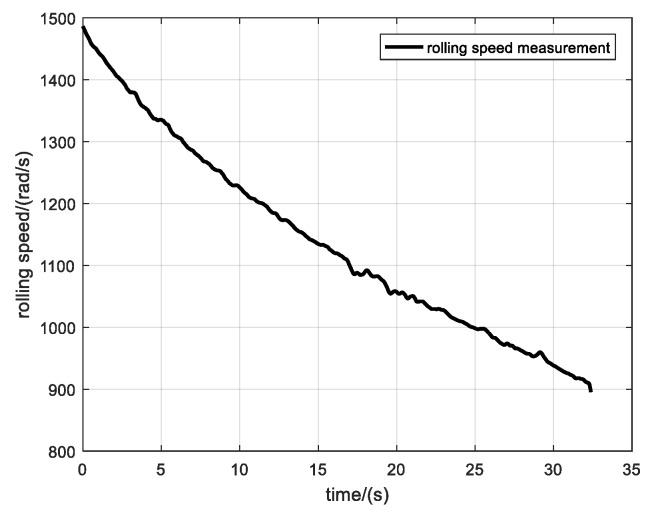
Rotational speed calculated using the zero-crossing method.

**Figure 16 sensors-19-05096-f016:**
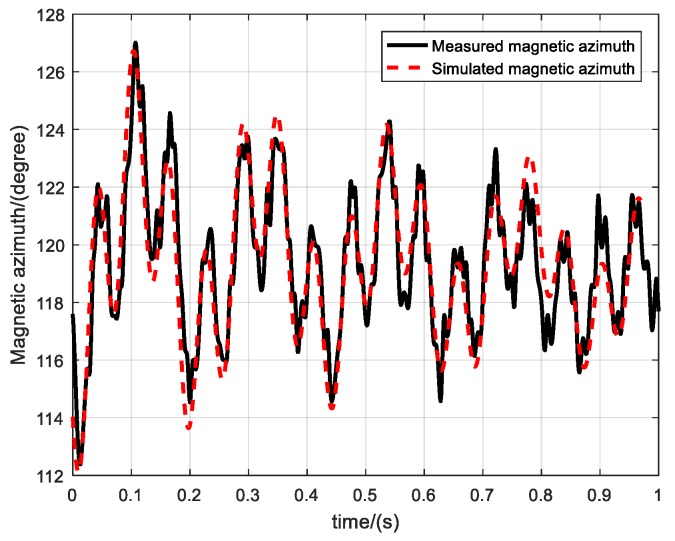
Initial alignment.

**Figure 17 sensors-19-05096-f017:**
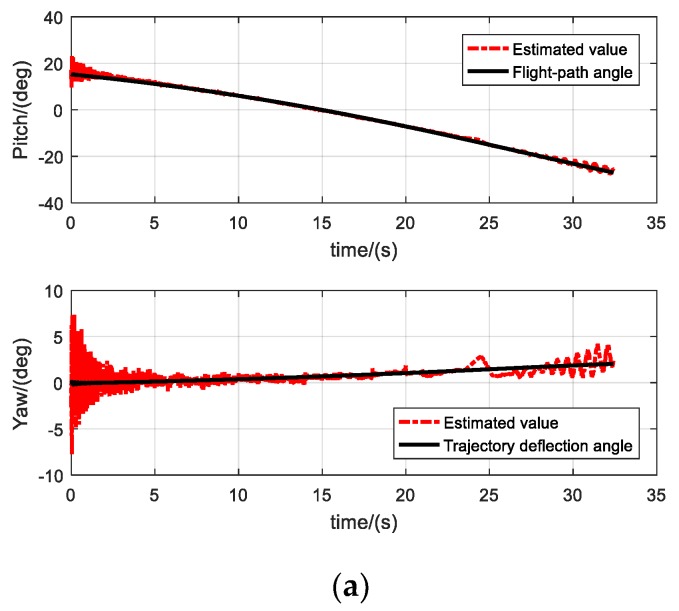
Pitch and yaw angles estimated by RK4-UKF. (**a**) Estimation of pitch and yaw of the entire ballistic, (**b**) Estimation of pitch angle in 2 s, (**c**) Calculation of yaw angle in 2 s.

**Figure 18 sensors-19-05096-f018:**
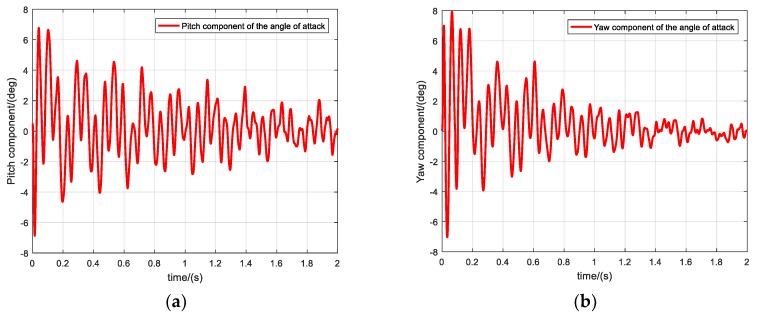
Pitch and yaw components of the attack angle. (**a**) Pitch component of the attack angle, (**b**) Yaw component of the attack angle.

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
