# Peer review of "A Novel Method for Estimating Pitch and Yaw of Rotating Projectiles Based on Dynamic Constraints"

_sensors, 2019, doi:10.3390/s19235096_

Round 1
Reviewer 1 Report
The paper presents a new methodology to estimate the attitude angles of a rotating projectile in a steady flight. The projectile dynamic equations are derived as a nonlinear state-space equation relating the pitch angle, yaw, and magnetic azimuth angle. The Unscented Kalman Filter (UKF) is applied to estimate the angle states, both with synthetic and experimental data. The studies show the feasibility of the zero-crossing method to characterize the geomagnetic azimuth. The dynamic characteristics of a high-spinning projectile in steady flight (magnetic declination and dip) is obtained with two different methods for attitude estimation. The paper is well written, with detailed derivations and through reference revision of the applied methods.
This reviewer suggests the authors review the citations of the references [15-21].
Author Response
To Sensors
Dear Reviewer,
Thank you for your letter and for the reviewers’ comments concerning our manuscript entitled “A Novel Method for Estimating Pitch and Yaw of Rotating Projectiles Based on Dynamic Constraints". Those comments are all valuable and very helpful for revising and improving our paper, as well as the important guiding significance to our researches. We have studied comments carefully and have made correction which we hope meet with approval. Revised portion are marked in red in the paper. The main corrections in the paper are as flowing:
We have revised the whole manuscript with the assistance from a colleague whose English is good and English teacher. Some terminology has been modified to ensure consistency with international terminology. The simulation of the entire ballistics was added, including the figures and the corresponding analysis. Added experimental processing of the entire ballistics, including the figures and the corresponding analysis. Added Monte Carlo simulation and analyzed the results. The sequence numbers of some references have been rearranged. Modified statements and words that were inaccurate in the manuscript; The figures containing the coordinate system were modified, and different coordinate systems were marked in different colors to make the coordinate system more recognizable.
The responds to the reviewer’s comments are as flowing:
Reviewer #1:
Response to comment: “The paper presents a new methodology to estimate the attitude angles of a rotating projectile in a steady flight. The projectile dynamic equations are derived as a nonlinear state-space equation relating the pitch angle, yaw, and magnetic azimuth angle. The Unscented Kalman Filter (UKF) is applied to estimate the angle states, both with synthetic and experimental data. The studies show the feasibility of the zero-crossing method to characterize the geomagnetic azimuth. The dynamic characteristics of a high-spinning projectile in steady flight (magnetic declination and dip) is obtained with two different methods for attitude estimation. The paper is well written, with detailed derivations and through reference revision of the applied methods. This reviewer suggests the authors review the citations of the references [15-21].”
Response 1: We are very grateful to you for your good comments, which is very encouraging.
The absence of reference [15-21] is a mistake made by the authors. They should have been placed in the position line 40: "The magnetometer can be widely used in attitude estimation of rotating objects [15–21] after undergoing a calibration and compensation process [12–14], thanks to its features of reliable Performance, low cost and no error accumulation."
Moreover, the manuscript was originally written in Chinese and then translated into English. Due to the different expressions between Chinese and English, the positions of references [12-14] and [15-21] were reversed. Therefore, the authors rearrange the references [12-14] and [15-21] at the references part of the manuscript and modify them in the text like this: "The magnetometer can be widely used in attitude estimation of rotating objects [12–18] after undergoing a calibration and Compensation process [19–21], thanks to its features of reliable performance, low cost and no error accumulation.”
We tried our best to improve the manuscript and made some changes in the manuscript. These changes will not influence the content and framework of the paper. And here we did not list the changes but marked in red in revised paper.
We appreciate for Editors/Reviewers’ warm work earnestly, and hope that the correction will meet with approval.
Once again, thank you very much for your comments and suggestions.
Sincerely Yours,
Liangliang An

Reviewer 2 Report
The flaws of this paper are the following:
The authors seem to never take into consideration the use of an IMU for estimating the attitude of the projectile. IMUs are the most common attitude sensor in many applications, including missiles. They can be cheap and small and are usually more accurate than magnetometers. The drift associated to gyroscopes can be compensated with adequate filtering techniques, including the UKF used by the authors. The authors affirm (line 306-7) that it is not possible to measure the attitude of a projectile, but it is not generally true. Many missiles have embedded IMUs that do this and their attitude estimation scheme is simpler. The simulation has been conducted on a single run. In order to analyze the behavior of the proposed estimation algorithm, it is necessary to conduct a Monte Carlo campaign of tests (> 100 runs), changing every time the noise on the measurements and the initial guess of the filter. The accuracy of the filter shall be analyzed by taking into account the standard deviation and the mean of the errors of the Monte Carlo runs. 1 sec of analysis seems a very short time to assess the behavior of the estimator The results of the experiment are not clear. Fig. 15 only show the estimated angles, but the true value or (even better) the estimation error is not shown. It is not possible to evaluate the behavior of the filter. The terminology used in this paper is sometimes unusual compared to that commonly used in the international community.
Author Response
To Sensors
Dear Reviewer,
Thank you for your letter and for the reviewers’ comments concerning our manuscript entitled “A Novel Method for Estimating Pitch and Yaw of Rotating Projectiles Based on Dynamic Constraints". Those comments are all valuable and very helpful for revising and improving our paper, as well as the important guiding significance to our researches. We have studied comments carefully and have made correction which we hope meet with approval. Revised portion are marked in red in the paper. The main corrections in the paper are as flowing:
We have revised the whole manuscript with the assistance from a colleague whose English is good and English teacher. Some terminology has been modified to ensure consistency with international terminology. The simulation of the entire ballistics was added, including the figures and the corresponding analysis. Added experimental processing of the entire ballistics, including the figures and the corresponding analysis. Added Monte Carlo simulation and analyzed the results. The sequence numbers of some references have been rearranged. Modified statements and words that were inaccurate in the manuscript; The figures containing the coordinate system were modified, and different coordinate systems were marked in different colors to make the coordinate system more recognizable.
The responds to the reviewer’s comments are as flowing:
Reviewer #2:
Response to comment: “The authors seem to never take into consideration the use of an IMU for estimating the attitude of the projectile. IMUs are the most common attitude sensor in many applications, including missiles. They can be cheap and small and are usually more accurate than magnetometers. The drift associated to gyroscopes can be compensated with adequate filtering techniques, including the UKF used by the authors.”
Response: Thank you for the comments about article. From your comments, we believe that you are a knowledgeable and experienced expert in this field, and we are very happy that you reviewed my manuscript.
The research involved in this manuscript was sponsored by a number of project funds. In fact, before the work started, we did consider using the IMU to measure the attitude of the projectiles. However, the high-precision IMU that can withstand high overload and measure extremely high spinning speed is too expensive to use on low-cost projectiles, especially when it will be used on a large scale in the future. The extremely high spinning speed of the projectile also causes the calibration of the gyroscope to be imperfect. Moreover, we hope to make some progress in the attitude measurement method of the projectiles using geomagnetic.
Response to comment: “The authors affirm (line 306-7) that it is not possible to measure the attitude of a projectile, but it is not generally true. Many missiles have embedded IMUs that do this and their attitude estimation scheme is simpler.”
Response: This mistake was caused by the translation of the manuscript. The original intention of the authors is: “At present, the technology cannot observe directly and record the flying attitude of a projectile with a range reaches tens of kilometers or even hundreds of kilometers.”
In the manuscript, this sentence is modified to: “It is impossible to observe directly and record the attitude of a flying projectile using current technology when the range reaches tens of kilometers or hundreds of kilometers.”
Response to comment: “The simulation has been conducted on a single run. In order to analyze the behavior of the proposed estimation algorithm, it is necessary to conduct a Monte Carlo campaign of tests (> 100 runs), changing every time the noise on the measurements and the initial guess of the filter. The accuracy of the filter shall be analyzed by taking into account the standard deviation and the mean of the errors of the Monte Carlo runs.”
Response: We must express our gratitude to the reviewer once again. The advice on performing Monte Carlo test is very helpful for the simulation verification, which makes the simulation more rigorous. We added a small chapter 5.1.2 to the manuscript, which specifically introduced the 1000 times Monte Carlo simulation, and plotted the figures to illustrate the results.
Response to comment: “1 sec of analysis seems a very short time to assess the behavior of the estimator. The results of the experiment are not clear.”
Response: Thank you for the good comment. In the simulation section, we added estimation results and estimation errors of the entire ballistic in Figures 7a, 7b. The analysis was carried out in the manuscript.
Figures 7a Figures 7b.
We also made the same changes to the experimental results.
Figures 17a
Response to comment: “ 15 only show the estimated angles, but the true value or (even better) the estimation error is not shown. It is not possible to evaluate the behavior of the filter.”
Response: Limited by the current attitude measurement technology and experimental conditions, the true value of the projectile attitude cannot be obtained in the field experiment, and the accuracy of the estimation cannot be quantified. The experiment is mainly to verify the feasibility of the method in practical engineering applications. The method is proved to be feasible and effective through the analysis of the flight stability of the projectile.
In the discussion of experimental results, we have added a paragraph at line 446 to briefly explain the purpose of the experiment and the focus of future work. We marked the changes in red.
Response to comment: “The terminology used in this paper is sometimes unusual compared to that commonly used in the international community.”
Response: The authors queried the relevant materials and literature, and changed some terminologies in the manuscript as follows:
â‘ : the angle between the velocity vector and the horizontal plane, is flight path angle;
: the angle between the velocity vector and the vertical plane, is trajectory deflection angle;
â‘¡ : ballistic coordinate system;
â‘¢ : elevation angle;
â‘£ : direction of fire;
⑤ ballistic deflection;
We made some other changes and marked in red in the manuscript.
We tried our best to improve the manuscript and made some changes in the manuscript. These changes will not influence the content and framework of the paper. And here we did not list the changes but marked in red in revised paper.
We appreciate for Editors/Reviewers’ warm work earnestly, and hope that the correction will meet with approval.
Once again, thank you very much for your comments and suggestions.
Sincerely Yours,
Liangliang An

Reviewer 3 Report
The paper is well written and organized. The results are interesting and well described, with sufficient experimental data. Few minor changes or modifications are suggested (see the attached file). Information about projectile dimensions or any possible restrictions of the described model will be useful. Future works and implementation can be added at the end of the paper.

Author Response
To Sensors
Dear Reviewer3,
Thank you for your letter and for the reviewers’ comments concerning our manuscript entitled “A Novel Method for Estimating Pitch and Yaw of Rotating Projectiles Based on Dynamic Constraints". Those comments are all valuable and very helpful for revising and improving our paper, as well as the important guiding significance to our researches. We have studied comments carefully and have made correction which we hope meet with approval. Revised portion are marked in red in the paper. The main corrections in the paper are as flowing:
We have revised the whole manuscript with the assistance from a colleague whose English is good and English teacher. Some terminology has been modified to ensure consistency with international terminology. The simulation of the entire ballistics was added, including the figures and the corresponding analysis. Added experimental processing of the entire ballistics, including the figures and the corresponding analysis. Added Monte Carlo simulation and analyzed the results. The sequence numbers of some references have been rearranged. Modified statements and words that were inaccurate in the manuscript; The figures containing the coordinate system were modified, and different coordinate systems were marked in different colors to make the coordinate system more recognizable.
The responds to the reviewer’s comments are as flowing:
Reviewer #3:
Response to comment: The English writing problems in the manuscript.
Response: Thank you very much for reviewing the manuscript carefully. There are three English writing problems in the manuscript, and we changed them as follows:
â‘´ line 77-79.
“It is necessary to discretize a continuous system when the filtering algorithm is applied in computers, and the discretization method and discrete step-size directly affect the filtering accuracy.” is changed to “It is necessary to discretize a continuous system when the filtering algorithm is applied in computers. The discretization method and discrete step-size directly affect the filtering accuracy.”
⑵ line 85.
“First” is changed to “first”.
â‘¶ line 309 and 311
Both “The” are changed to “the”.
Response to comment: “Line 251-3:“Assume that the projectile is launched from a location of (E100°,N39° ) with an initial velocity of 800m/s, a shooting angle of 60° and a shooting direction of 100°.”Why you are assuming this value? References?”
Response: This part of the manuscript is a simulation. These assumptions are the initial settings of the simulation. These values are necessary for the simulation data, but not necessary for the analysis of the simulation results. The reason why these simulation initial settings are written here is mainly to increase the credibility of the simulation experiment, and also to facilitate later researchers to reproduce the simulation. These initial assumptions may not be written if the reviewer insists.
Response to comment: “Line 317:“The launch angle was 15.3° and the shooting direction was 103.3555°.”Maybe a picture of the setting can explain better the experiment configuration.”
Response: We agree with your suggestion. The authors have added a figure of the experimental layout to the manuscript and reset the serial numbers of the other figures. The figure is like this:
Figures 10
Response to comment: “Information about projectile dimensions or any possible restrictions of the described model will be useful.”
Response: We agree with you. We deleted all information about the projectile. Thank you again for the suggestion.
Response to comment: “Future works and implementation can be added at the end of the paper.”
Response: In the conclusion at line 468, we have added a paragraph to briefly introduce the direction of future work, and marked in red.
We tried our best to improve the manuscript and made some changes in the manuscript. These changes will not influence the content and framework of the paper. And here we did not list the changes but marked in red in revised paper.
We appreciate for Editors/Reviewers’ warm work earnestly, and hope that the correction will meet with approval.
Once again, thank you very much for your comments and suggestions.
Sincerely Yours,
Liangliang An
Round 2
Reviewer 2 Report
Thank you for accepting my comments